# Contrast, Attend and Diffuse to Decode High-Resolution Images from Brain Activities

**Jingyuan Sun†‡**
KU Leuven
jingyuan.sun@kuleuven.be

**Mingxiao Li†**
KU Leuven
mingxiao.li@kuleuven.be

**Zijiao Chen**
National University of Singapore
zijiao.chen@u.nus.edu

**Yunhao Zhang**
Chinese Academy of Science
zhangyunhao2021@ia.ac.cn

**Shaonan Wang**
Chinese Academy of Science
shaonan.wang@nlpr.ia.ac.cn

**Marie-Francine Moens**
KU Leuven
sien.moens@kuleuven.be

## Abstract

Decoding visual stimuli from neural responses recorded by functional Magnetic Resonance Imaging (fMRI) presents an intriguing intersection between cognitive neuroscience and machine learning, promising advancements in understanding human visual perception. However, the task is challenging due to the noisy nature of fMRI signals and the intricate pattern of brain visual representations. To mitigate these challenges, we introduce a two-phase fMRI representation learning framework. The first phase pre-trains an fMRI feature learner with a proposed Double-contrastive Mask Auto-encoder to learn denoised representations. The second phase tunes the feature learner to attend to neural activation patterns most informative for visual reconstruction with guidance from an image auto-encoder. The optimized fMRI feature learner then conditions a latent diffusion model to reconstruct image stimuli from brain activities. Experimental results demonstrate our model's superiority in generating high-resolution and semantically accurate images, substantially exceeding previous state-of-the-art methods by $39.34\%$ in the 50-way-top-1 semantic classification accuracy. The code implementations will be available at `https://github.com/soinx0629/vis_dec_neurips/`.

## 1 Introduction

Reconstructing visual stimuli from neural imaging data represents a promising interdisciplinary area between cognitive neuroscience and machine learning [1]. A system capable of accurately decoding neural responses to visual input can help illuminate the complex mechanisms underlying the brain's visual perception [2, 3]. Furthermore, it can elucidate the relationships between human visual systems and computational vision models [4, 5, 6]. Such a system also holds the potential to assist patients, particularly those with motor disabilities, in expressing their thoughts and intentions through brain signals.

Functional Magnetic Resonance Imaging (fMRI), as a non-invasive method to measure brain activity, has been extensively utilized to decipher perceptions from neural responses [5, 7, 8]. However,

---

†Equal Contribution
‡Corresponding Author

37th Conference on Neural Information Processing Systems (NeurIPS 2023).

despite its utility, achieving reliable and robust visual reconstructions from fMRI recordings remains a significant challenge [9]. Primarily, the fMRI data is inherently noisy. The recorded signals encompass not only the specific responses elicited by visual stimuli but also incorporate additional sources of noise arising from various cognitive, physiological processes, and scanner operations [10, 11]. These noises can obscure the neural activation patterns associated with the stimulus, thereby complicating the direct decoding of visual information from the fMRI data. Moreover, the process by which a visual stimulus arouses a neural response is dynamic, intricate and multifaceted. It involves multiple stages of neural processing [12, 13], from the initial perception in the retina to higher-order cognitive operations within the brain's visual and associative regions [14, 15]. The resulting fMRI signal is a highly convolved representation of these distinct processing stages, rather than a linear one [16]. It is thus non-trivial to reverse this process and disentangle the convolved representations to achieve visual decoding.

Existing methods for vision decoding tend to struggle with such challenging complexity. Previous pioneering work has relied on traditional statistical approaches, such as ridge regression, to map fMRI signals back to the corresponding stimuli [17, 18]. However, these oversimplified methods often fall short of capturing the non-linear relationship between the stimulus and the neural responses. More recently, deep learning methods such as Generative Adversarial Networks (GAN) [9, 19] and latent diffusion models (LDMs) [7, 5] have been adopted to model the non-linearity and yield better results. But the difficulty of disentangling vision-related brain activities from noises still hinders these methods from decoding images with optimal accuracy.

To navigate these challenges, we propose a double-phase fMRI representation learning framework. In Phase 1, we pre-train the fMRI feature learner on large-scale unlabeled fMRI data with a novel Double-contrastive Masked Auto-encoder (DC-MAE). The DC-MAE helps discern common patterns of brain activities shared among populations over individual noises. In Phase 2, we further tune the feature learner on the fMRI-image parallel data with an image auto-encoder. The pixel-level guidance from the image auto-encoder teaches the fMRI auto-encoder to attend to brain signals that are most informative for image reconstruction. The trained fMRI feature learner is then used to condition an LDM to reconstruct image stimuli from brain activities. Experimental results demonstrate that the proposed model generates high-resolution and semantically accurate images, outperforming the previous state-of-the-art method by $39.34\%$ in 50-way-top-1 accuracy. Our research paves the way for further exploration of the potential of decoding tasks.

## 2 Related Works

### 2.1 Visual Decoding from fMRI

Driven by the substantial potential, recent years have witnessed a growing interest in reconstructing visual experiences from fMRI data. This task has been examined in various contexts, including explicitly presented visual stimuli [4, 20], perceived emotions [21] and even imagined content [1]. Though studies on this task keep emerging, the challenges presented by the low signal-to-noise ratio (SNR) of fMRI data and the high complexity of brain visual representations still exist. In the initial stage in this field, efforts to identify or reconstruct visual images from fMRI primarily utilized handcrafted features [17] and traditional regression models [18, 22]. Nonetheless, oversimplified methods can not fully account for the intricate patterns of brain visual representations, generating blurry and semantically meaningless images.

Recent research has shifted towards artificial neural network representations and deep generative models [23]. Typically, such models mapped fMRI signals to image features and fine-tuned pre-trained generative models like Generative Adversarial Networks (GANs) [9, 19] or Diffusion Models [7, 24, 25, 26, 27] to generate images from the mapped features. For example, [28] decoded fMRI data to hierarchical image features extracted by a pre-trained VGG and fed the predicted features to a GAN. But [28], like other parallel work [29, 1], used fMRI directly for training and decoding. Without explicitly denoising the fMRI representations, though these works outperformed traditional regression-based models, they still fell short by generating implausible images. In contrast, our framework contains an individual phase to learn denoised fMRI representations with DC-MAE. [7] adopted a naive MAE to learn fMRI representations. But compared with [7], our framework further introduces pixel-level guidance from the image auto-encoder to help disentangle the vision-related neural activities from background noises. Our model is also not similar to other works [30, 31] which

directly map fMRI to the image feature space and inevitably face data incompatibility between the two very different modalities. We encourage the emergence of a cross-modality representation space by guiding the fMRI auto-encoder with a pre-trained image auto-encoder. Experimental results prove the superiority of our methods over these related baselines.

## 2.2 Diffusion Probabilistic Models

Diffusion probabilistic models have been empirically established as powerful generative models for image generation surpassing GANs [32] in terms of both diversity and fidelity. The diffusion model was first proposed in [33] and further improved by [26, 34]. The quality of generated image could also be enhanced by training-free methods [35, 36]. The initial diffusion models were primarily applied in the pixel space. They achieve impressive success in generating images of high quality, but also suffer from significant drawbacks such as prolonged inference time and substantial training costs [26]. To address these two issues, [27] introduce the latent diffusion model (LDM), also referred to as stable diffusion (SD). This approach adopts a pretrained Vector Quantized Generative Adversarial Network (VQGAN) [37] or Variational auto-encoder (VAE) [38] to construct a latent image space, within which optimization and evaluation are performed. The LDM not only generates images of high quality, but also alleviates the computational burden. Moreover, the incorporation of a cross-attention mechanism within the attention block of the diffusion UNet model permits the LDM model to offer a broad spectrum of controls in image synthesis. This includes textual controls [39, 40, 41], controls over images in various domains [42, 43] such as depth maps, sketch maps, or candy maps. Such versatility and adaptability give the LDM model a substantial advancement in the field of image synthesis.

## 3 Methods

### 3.1 Motivation and Overview

In this subsection, we provide a concise analysis of the fMRI data and brain visual representations that motivate the design of our method.

First, the fMRI recordings are inherently noisy, subject to various sources of physiological and scanner-related noises [44]. FMRI records not only responses to visual stimuli but also signals from other cognitive processes. Second, fMRI quantifies changes in the blood-oxygen-level-dependent (BOLD) signal. Adjacent voxels are often found to display similar magnitudes, suggesting the spatial redundancy of fMRI data [2]. Third, neural responses to identical stimuli can exhibit significant divergence [45, 46] across populations.

Considering these analyses altogether, we propose a double-phase fMRI representation learning framework. In Phase 1, we pre-train an MAE with a contrastive loss to learn fMRI representations from unlabeled data. The masking which sets a certain portion of the input data to zero targets the spatial redundancy of fMRI data. The calculation of recovering the original data from the remaining after masking suppresses noises. Optimization of the contrastive loss discerns common patterns of brain activities over individual variances. After pre-training in Phase 1, we tune the fMRI auto-encoder with an image auto-encoder. We expect the pixel-level guidance from the image auto-encoder to support the fMRI auto-encoder in disentangling and attending to brain signals related to vision processing. After FRL Phase 1 and Phase 2, we apply the representation learned by the fMRI auto-encoder as conditions to tune the LDM and generate the image from the brain activities.

### 3.2 fMRI Representation Learning (FRL)

**Phase 1: Pre-training Double-Contrastive Masked Auto-Encoder (DC-MAE)**  We introduce a method termed the "Double-Contrastive Masked Auto-Encoder". DC-MAE has been specifically designed to pre-train fMRI representations from unlabeled data, as inspired by previous work in visual contrastive learning [47]. As shown in Figure 1, the DC-MAE consists of an encoder $E_F$ and a decoder $D_F$. $E_F$ takes a masked version of the fMRI signal as the input, and $D_F$ is trained to predict the unmasked fMRI. The term "Double-Contrastive" implies that the model optimizes contrastive losses and engages in two separate contrasting processes during the representation learning of an fMRI example.

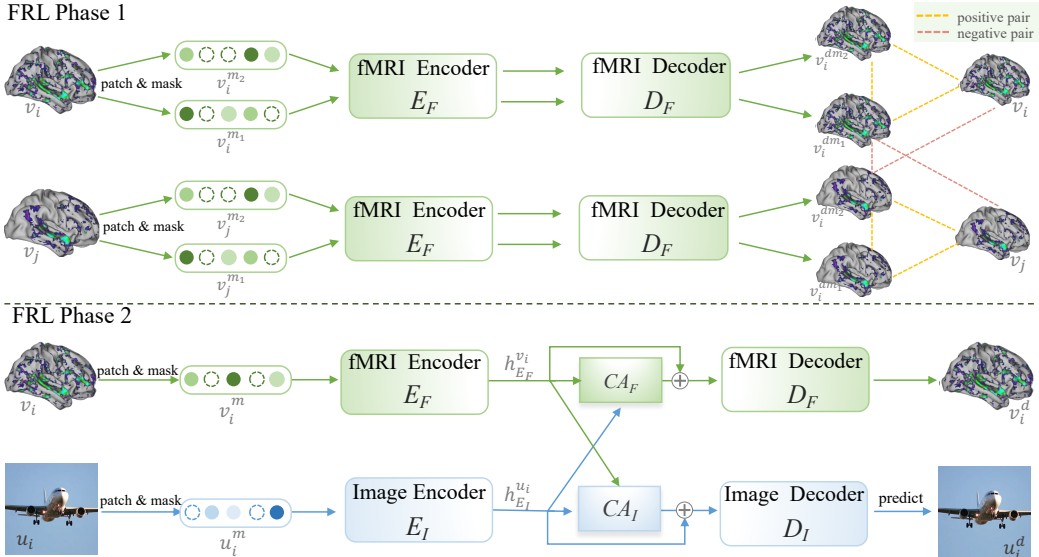

Figure 1: The Phase 1 and Phase 2 of the fMRI Representation Learning framework. After the fMRI feature learner is pre-trained in Phase 1, it will be tuned with an image auto-encoder in Phase 2. The definitions of depicted variables in this figure are detailed in Section 3.1.

Specifically, in the first contrasting, each sample $v_i (i \leq n)^1$ within a batch of $n$ fMRI examples $v$ undergoes two separate instances of random masking. This procedure yields two distinct masked versions of $v_i$, referred to as $v_i^{m_1}$ and $v_i^{m_2}$. These masked versions establish a positive sample pair for the first contrasting. Subsequently, a 1D convolutional layer, characterized by a stride that matches the patch size, tokenizes $v_i^{m_1}$ and $v_i^{m_2}$ into embeddings. These embeddings are then fed independently into the same fMRI encoder $E_F$. The decoder $D_F$, in turn, receives each of the encoded latent representations as input and generates predictions $v_i^{dm_1}$ and $v_i^{dm_2}$. The first contrastive loss, denoted as the cross-contrastive loss, is thus calculated through an InfoNCE loss [48] as follows:

$$\mathcal{L}_C = -\log \frac{\exp\left(v_i^{dm_1} \cdot v_i^{dm_2}/\tau\right)}{\exp\left(v_i^{dm_1} \cdot v_i^{dm_2}/\tau\right) + \sum_{j \neq i} \exp\left(v_i^{dm_1} \cdot v_j^{dm_1}/\tau\right)} \tag{1}$$

In the second contrasting, an unmasked original image $v_i (i \leq n)$ and its corresponding masked image $v_i^m$ form an inherent positive sample pair. Here, $v_i^{dm}$ denotes the predicted image output from the decoder $D_F$. The second contrastive loss, referred to as the self-contrastive loss, is computed through as:

$$\mathcal{L}_S = -\log \frac{\exp\left(v_i^{dm} \cdot v_i/\tau\right)}{\exp\left(v_i^{dm} \cdot v_i/\tau\right) + \sum_{j \neq i} \exp\left(v_i^{dm} \cdot v_j^{dm}/\tau\right)}. \tag{2}$$

Optimizing the self-contrastive loss $\mathcal{L}_S$ can also achieve mask-reconstruction. For both $\mathcal{L}_C$ and $\mathcal{L}_S$, the negative examples $v_j (j \neq i)$ are sourced from the same batch of instances. $\mathcal{L}_C$ and $\mathcal{L}_S$ are optimized jointly as follows:

$$\mathcal{L} = \gamma_C \mathcal{L}_C + \gamma_S \mathcal{L}_S \tag{3}$$

In this equation, the hyper-parameters $\gamma_C$ and $\gamma_S$ regulate the weight of each loss.

---

[1]Note that this is a 1-dimensional vector, not a time series, as we have averaged the data across the time dimension. This results in a spatial pattern of fMRI signal over the visual cortex for each picture viewed by the subject. We then employ a 1D convolutional model to transform this 1D spatial pattern of the fMRI signal, $v_i$, into an embedding.

**Phase 2: Tuning with Cross Modality Guidance** Considering the relatively low signal-to-noise ratio (SNR) and highly convolved nature of fMRI recordings , it is important for the fMRI feature learner to attend to the brain activation patterns most relevant for visual processing and most informative for reconstruction.

So as shown in Figure 1, after pre-training in Phase 1, the fMRI auto-encoder is tuned to reconstruct fMRI with the aid of an MAE for image, and vice versa in Phase 2. Specifically, denote one image from a batch of $n$ samples as $u_i (i \leq n)$ and the fMRI recorded neural responses to $u_i$ as $v_i$. $u_i$ and $v_i$ go through patching and random masking to be $u_i^m$ and $v_i^m$. $u_i^m$ and $v_i^m$ are respectively fed in the image encoder $E_I$ and fMRI encoder $E_F$, getting $h_{E_I}^{u_i} = E_I(u_i^m)$ and $h_{E_F}^{v_i} = E_F(v_i^m)$. For reconstructing the fMRI $v_i$, $h_{E_I}^{u_i}$ and $h_{E_F}^{v_i}$ are merged with a cross-attention module as follows:

$$Q_I^u = W^{Q_I} h_{E_I}^{u_i} + b^{Q_I}; K_F^v = W^{K_F} h_{E_F}^{v_i} + b^{K_F}; V_F^{v_i} = W^{V_F} h_{E_F}^{v_i} + b^{V_F}$$

$$CA_F(Q_I^{u_i}, K_F^{v_i}, V_F^{v_i}) = softmax(\frac{Q_I^{u_i}(K_F^{v_i})^T}{\sqrt{d_k}})V_F^{v_i} \tag{4}$$

$W$ and $b$ denote the weights and biases of corresponding linear layers. $\sqrt{d_k}$ is the scaling factor and $d_k$ is the dimension of the key vectors. $CA$ is the abbreviation of cross-attention. $CA_F(Q_I^{u_i}, K_F^{v_i}, V_F^{v_i})$ is then added to $h_{E_F}^{v_i}$ and fed into the fMRI decoder to reconstruct $v_i$ as $v_i^d$.

$$v_i^d = D_F(h_{E_F}^{v_i} + CA_F(Q_I^{u_i}, K_F^{v_i}, V_F^{v_i})) \tag{5}$$

A similar computation is also conducted in the image auto-encoder. The output $h_{E_I}^{u_i}$ of the image encoder $E_I$ is merged with the output of $E_F$ through a cross-attention module $CA_I$, and then used to decode an image $u_i$ as $u_i^d$:

$$u_i^d = D_I(h_{E_I}^{u_i} + CA_I(Q_F^v, K_I^{u_i}, V_I^{u_i})) \tag{6}$$

The fMRI and image auto-encoders are trained jointly by optimizing the following loss:

$$L = \gamma_F(v_i - v_i^d)^2 + \gamma_I(u_i - u_i^d)^2 \tag{7}$$

### 3.3 Image Generation with Latent Diffusion Model (LDM)

After the fMRI feature learner is trained through FRL Phase 1 and Phase 2, we use its encoder $E_F$ to condition a LDM to generate images from brain activities. As displayed in Figure 2, the diffusion model consists of a forward diffusion process and a reverse denoising process. The forward process gradually degrades an image to a normal Gaussian noise by incrementally introducing Gaussian noise of variable variance. This process can be mathematically formulated as $q(x_t|x_{t-1}) = \mathcal{N}(x_t, \sqrt{\alpha}x_{t-1}, (1 - \alpha_t)I)$, where $t$ denotes the temporal step and $\alpha$ encompasses predefined noise schedule parameters. During the training phase, the diffusion model, specifically the U-Net [49] model, is optimized with the loss function below to learn the noise $\epsilon_t$ added at each time step in the forward process.

$$\mathcal{L}_t^{simple} = E_{t,x_0,\epsilon_t \sim \mathcal{N}(0,1)}[\|\epsilon_t - \epsilon_\theta(z_t, t)\|_2^2] \tag{8}$$

In the backward process, an image is synthesized by progressively eliminating noise from a randomly initialized standard Gaussian noise. The Latent Diffusion Model (LDM) [27] performs both forward and backward processes in a low-dimensional latent space which is constructed using a pre-trained VQ-VAE model [38]. This approach significantly mitigates the computational complexity while simultaneously maintaining the generated image's quality. We leverage the LDM as the backbone of our image generation model.

Decoding fMRI to a natural image can be seen as conditional image generation task. Given the relatively low SNR inherent in fMRI, combined with the limited quantity of fMRI-to-image data pairs, training an fMRI-to-image generation model from scratch presents substantial challenges. Consequently, the objective of this phase is to harness the fMRI to extract image-related knowledge from a pre-existing conditional image generation model.

Our approach extracts visual knowledge from the pretrained label-to-image LDM to generate image with the conditioning of fMRI. We incorporate the fMRI information into the LDM via a cross-attention operation similar to equation (4), as proposed in [27]. To further enhance the guidance provided by the conditional information, following the methodology of previous research [7], we

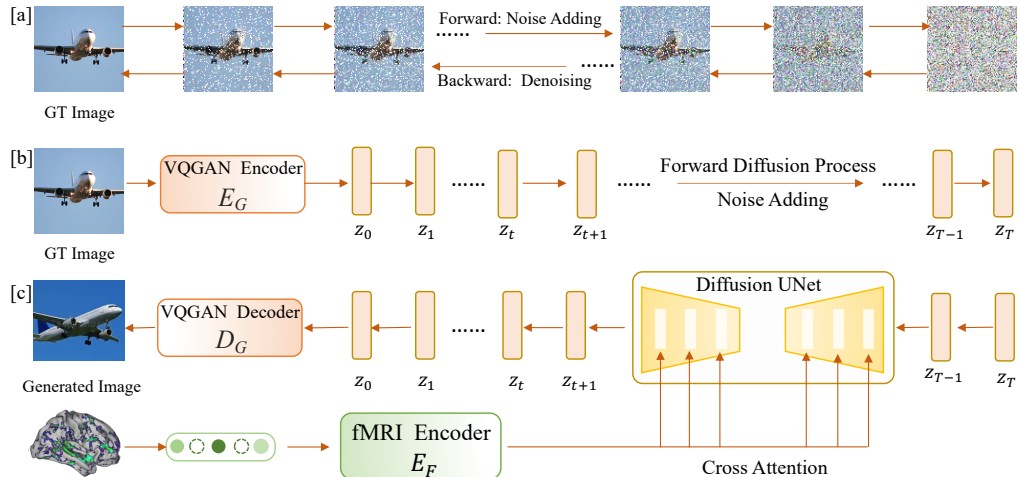

Figure 2: [a] Demo of the forward and backward processes of the diffusion model. [b] The forward process of the diffusion model which progressively corrupts an image with Gaussian noise. [c] In the backward process, the diffusion model, conditioned on our pretrained fMRI encoder, gradually denoises white noises to generate the image.

implement both cross-attention conditioning and time steps conditioning [50]. During training, given an image $u$ and fMRI $v$, along with VQGAN encoder $E_G$ and the fMRI encoder $E_F$ (which is trained in FRL Phase 1 and 2), we freeze the LDM and finetune the fMRI encoder using the following loss:

$$\mathcal{L}_t^{simple} = E_{t,u_0,\epsilon_t \sim \mathcal{N}(0,1)}[\|\epsilon_t - \epsilon_\theta(\phi(E_G(u))_t, t, E_F(v))\|_2^2] \tag{9}$$

Here, $\phi(E_g(u))_t = \sqrt{\overline{\alpha}_t}E_g(u) + \sqrt{1 - \overline{\alpha}_t}\epsilon$ and $\overline{\alpha}_t$ is the noise schedule of diffusion model. During the inference phase, the process begins with a standard Gaussian noise at time step $T$. The LDM follows the backward process sequentially to progressively denoise the hidden representation, conditioning on the given fMRI information. Upon reaching time step zero, the VQGAN decoder $D_G$ is used to convert the hidden representations to an image. The forward and backward process of the LDM are detailed in Figure 2[b,c].

## 4 Experimental Setup

### 4.1 fMRI Datasets

**HCP** The Human Connectome Project (HCP) originally serves as an extensive exploration into the connectivity of the human brain. It offers an open-sourced database of neuroimaging and behavioral data collected from 1,200 healthy young adults within the age range of 22-35 years. Currently, it stands as the largest public resource of MRI data pertaining to the human brain, providing an excellent foundation for the pre-training of brain activation pattern representations. Of the subjects involved, 1113 underwent scanning via a Siemens Skyra Connectom scanner for 3T MR, while a Siemens Magnetom scanner for 7T MR was utilized for the remaining 184. For the scope of this paper, we will predominantly focus on the data derived from the more populated 3T dataset.

**GOD** The Generic Object Decoding (GOD) Dataset is a specialized resource developed for fMRI-based decoding. It aggregates fMRI data gathered through the presentation of images from 200 representative object categories, originating from the 2011 fall release of ImageNet. The training session incorporated 1,200 images (8 per category from 150 distinct object categories). In contrast, the test session included 50 images (one from each of the 50 object categories). It is noteworthy that the categories in the test session were unique from those in the training session and were introduced in a randomized sequence across runs. On five subjects the fMRI scanning was conducted.

**BOLD5000** The BOLD5000 dataset is a result of an extensive slow event-related human brain fMRI study. It comprises 5,254 images, with 4,916 of them being unique. This makes it one of the most comprehensive publicly available datasets in the field. The dataset's principal advantage is its high

diversity, enabling the capture of the complexity and variability inherent in natural visual stimuli. The images in BOLD5000 were selected from three popular computer vision datasets: ImageNet, COCO, and Scenes. ImageNet provided 1,916 images primarily focusing on singular objects. Meanwhile, COCO contributed 2000 images featuring multiple objects, and Scenes contributed 1000 images depicting hand-crafted indoor and outdoor scenes. Four participants labeled CSI1 through CSI4, were involved in this study and underwent scanning via a 3T Siemens Verio MR scanner equipped with a 32-channel phased array head coil.

## 4.2 Implementation Details

### 4.2.1 FMRI Representation Learning (FRL)

For both FRL Phase 1 and Phase 2, the fMRI auto-encoder is the same ViT-based masked auto-encoder (MAE). We employed an asymmetric architecture for the fMRI auto-encoder, in which the decoder is considerably smaller with 8 layers than the encoder with 24 layers. We used a larger embedding-to-patch size ratio, specifically a patch size of 16 and an embedding dimension of 1024 for our model. We used random sparsification (RS) as a form of data augmentation, randomly selecting and setting 20% of voxels in each fMRI to zero.

**FRL Phase 1**  In Phase 1, we train the masked ViT-based fMRI auto-encoder with contrastive loss. For GOD subject 1,4,5 and BOLD5000 CSI 1,2, self-contrastive ($\gamma_s$) and cross-contrastive ($\gamma_c$) loss weights are both 1. The masking ratio is 0.5. For GOD subject 2,3 and BOLD5000 CSI 3,4, $\gamma_s = 1$ and $\gamma_c = 0.5$, masking ratio is 0.75. We set the batch size to 250 and train for 140 epochs on one NVIDIA A100 GPU. We train with 20-epoch warming up and an initial learning rate of 2.5e-4. We optimize with AdamW and weight decay 0.05.

**FRL Phase 2**  In Phase 2, we tune the fMRI autoencoder jointly with an image auto-encoder, which is a pre-trained ViT-based MAE released by [51]. The image auto-encoder has a 12-layer encoder with a 768 hidden size and a 6-layer decoder with a 512 hidden size. We set the batch size to be 16 and train for 60 epochs. We train with 2-epoch warming up. The initial learning rate is 5.3e-5. We optimize with AdamW and weight decay 0.05. We freeze the parameters of the decoder of the image-autoencoder and only tune the encoder. The two phases in total take about 12 hours to run. After the two phases, we only save the checkpoint of the fMRI encoder which has 15.16M parameters

### 4.2.2 Fine-tuning LDM

In this stage, we jointly optimize the parameters of LDM cross-attention heads and the fMRI encoder, while keeping other parameters of LDM unchanged. Given an fMRI-image pair, we first use the pre-trained VQ encoder to encode the image to obtain the latent representation which is further used as an objective to guide the joint training of the fMRI encoder and LDM cross-attention heads. During training, the fMRI data passes through the fMRI encoder trained using FRL, producing a patchified representation. This representation is then projected into key and value representation of cross-attention modules in the UNet of LDM. Furthermore, it is added to the time embedding to conduct double conditioning. The training follows the regular training pipeline of the diffusion model, where the model is optimized to learn to predict the Gaussian noise added to the image latent representation at each time step with the guidance of the given conditioning information. Here, we use the output of the fMRI encoder as the conditioning information. We conduct training with the following parameters: the batch size of 5, diffusion steps of 1000, the AdamW optimizer, a learning rate of $5.5e - 5$, and an image resolution of $256 \times 256 \times 3$.

## 4.3 Baseline Models and Evaluation Metric

**Baseline Models**  We juxtapose our proposed model with recently published benchmarks: IC-GAN [52], Self-supervised auto-encoder (SS-AE) [53], and DC-LDM [7]. The IC-GAN is based on an Instance-Conditioned GAN, whereas the SS-AE utilizes cycle consistency and perceptual losses to reconstruct images from fMRI brain recordings. The DC-LDM employs a double-conditioned LDM, demonstrating superior performance on the GOD and BOLD5000 datasets. These benchmarks reflect prevalent methodologies in visual reconstruction.

**Evaluation Metric**  Visual decoding prioritizes semantic consistency. Therefore, our evaluation metric is the $n$-way top-$k$ accuracy, which aligns with the precedent set in the literature [53, 7]. We

employ the pre-trained ImageNet-1K classifier [54] as a semantic correctness evaluator. During the evaluation, both the generated image and the corresponding ground truth image are fed into the classifier. Semantic correctness is then determined based on whether the top-$k$ classification among $n$ randomly selected classes aligns with the ground-truth classification. Further details can be referenced in the appendix.

## 5 Results

### 5.1 Reconstruction Results

In this section, we present a comparative analysis of our results with preceding studies, which include DC-LDM, IC-GAN, and SS-AE. We have evaluated our model on both GOD and BOLD5000 datasets. Following the setting of previous work [7], we first display in detail the results in Figure 3 for GOD subject 3 and BOLD5000 subject CSI 1 which achieve the best performance in their respective dataset. It is noteworthy that the original DC-LDM implementation utilized test set fMRI data for tuning, a setting not adopted by IC-GAN and SS-AE. To maintain a fair comparison, we first prohibit the use of test set fMRI data by the DC-LDM. As shown in Figure 3[a], our method surpasses the previous models by a large margin. For instance, our model achieves an accuracy exceeding that of DC-LDM and IC-GAN by over $39.34\%$ (calculated by $(25.080-17.999)/25.080 \times 100\% \approx 39.34\%$, comparing our model's accuracy $25.080$ against the DC-lDM's accuracy $17.999$) and $66.7\%$ respectively. The improvement of our model over the baselines is significant. All the significant results have p-value < 0.01 with paired t-tests.

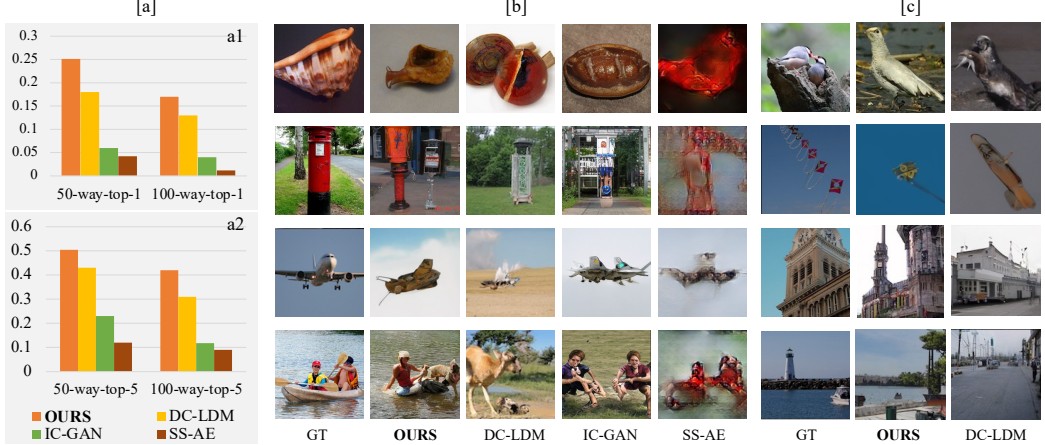

Figure 3: Reconstruction results. [a] Top-1 (a1) and top-5 (a2) classification accuracy of our model and other baselines on GOD subject 3. [b] Samples of reconstructed images and their ground truth from GOD subject 3's data. [c] Samples of reconstructed images and their ground truth from the BOLD5000 CSI 1's data.

To further investigate the quality of the images produced by different models, we randomly select 5 samples from the GOD test set and exhibit the generated images in Figure 3[b]. The SS-AE model can only generate a general layout, while the remaining three models are cable of producing images with a semantic meaning similar to that of ground truth image. Our model generates images of superior quality that exhibits a higher degree of semantic consistency with the ground truth images. In Figure 3[c], we compare images generated by our model and DC-LDM. (IC-GAN and SS-AE were not trained on BOLD5000 in their original paper). Our model achieves 50-way-top-1 accuracy of 25 on CSI1. Though both models can produce high-quality images, the image generated by our model bears a semantic meaning more consistent with the ground truth images.

To check if our model may reliably reconstruct brain activities on different subjects, we further evaluate it against DC-LDM on all the other 4 subjects of the GOD dataset. The bar charts in Figure 4[a] below show the results in 50-way-top-1 classification accuracy. Our model substantially outperforms the previous state-of-the-art method (DC-LDM) on all GOD subjects. To achieve DC-LDM's reported performance in its original paper [7], this method need signals from test set fMRI

data. To ensure a fair evaluation, we banned DC-LDM from tuning on the test set in the comparison of the Figure 3. But we show here that, our model still largely exceeds DC-LDM on four GOD subjects even after DC-LDM is tuned on the test set fMRI data.

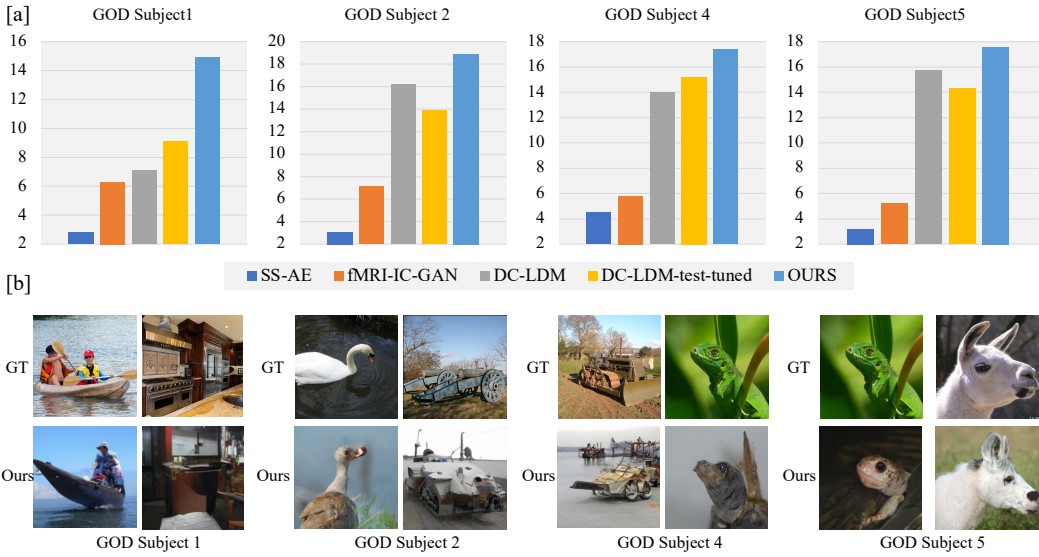

Figure 4: Reconstruction performance of our model and other baselines on GOD subjects 1, 2, 4 and 5, measured by 50-way-top-1 classification accuracy

## 5.2 Ablation Study Results

We train the FRL Phase with different settings of hyper-parameters. The trained fMRI encoders then guide the LDM in generating images. We use the 50-way-top-1 accuracy of LDM image generation to measure the influence of hyper-parameter values. Due to space limit, in this section we mainly focus on FRL Phase 2 where direct interaction between the image and fMRI data takes place which may more highly influence the final reconstruction performance.

| Ablation parameter | ID | fMRI rec. loss weight | Image rec. loss weight | fMRI mask ratio | fMRI mask ratio | Dec. layer num | Accuracy |
|---|---|---|---|---|---|---|---|
| fMRI and image reconstruction loss weight | 1 | 0 | 1 | 0.75 | 0.75 | 6 | 12.52 |
| | 2 | 1 | 0 | 0.75 | 0.75 | 6 | 16.64 |
| | 3 | 0.5 | 1.5 | 0.75 | 0.75 | 6 | 17.68 |
| | 4 | 1 | 1 | 0.75 | 0.75 | 6 | 19.64 |
| | 5 | 1.5 | 0.5 | 0.75 | 0.75 | 6 | 19.76 |
| | 6 | 0.25 | 1.5 | 0.75 | 0.75 | 6 | 20.8 |
| Ablation parameter | ID | fMRI rec. loss weight | Image rec. loss weight | fMRI mask ratio | Image mask ratio | Dec. layer num | Accuracy |
| fMRI and image mask ratio | 7 | 1 | 1 | 0.5 | 0.75 | 6 | 15.52 |
| | 8 | 1 | 1 | 0.5 | 0.5 | 6 | 17.72 |
| | 9 | 1 | 1 | 0.5 | 0.25 | 6 | 19 |
| | 4 | 1 | 1 | 0.75 | 0.75 | 6 | 19.64 |
| | 10 | 1 | 1 | 0.75 | 0.5 | 6 | 21.04 |
| Ablation parameter | ID | fMRI rec. loss weight | Image rec. loss weight | fMRI mask ratio | Image mask ratio | Dec. layer num | Accuracy |
| number of fMRI and image decoder layer | 11 | 1 | 1 | 0.75 | 0.5 | 2 | 17.44 |
| | 12 | 1 | 1 | 0.75 | 0.5 | 4 | 17.76 |
| | 4 | 1 | 1 | 0.75 | 0.5 | 6 | 19.64 |
| | 13 | 1 | 1 | 0.75 | 0.5 | 8 | 17.44 |
| Best parameter | 14 | 0.25 | 1.5 | 0.75 | 0.5 | 6 | 25.08 |

Table 2: Ablation study of FRL Phase 2 on GOD subject 3. Cells with colored shades denote the hyper-parameters to be tuned in one ablation group. For example, cells with yellow shades denote that fMRI and image mask ratio are the parameters to be tuned while other parameters are kept the same.

**Reconstruction Loss Weight** In the FRL Phase 2, the feature learner is tuned by optimizing the joint loss of fMRI and image reconstruction, as in equation (7). So we first focus on the influence of the fMRI and image reconstruction loss weights, namely $\gamma_F$ and $\gamma_i$. The results are reported in Table 2 (ID1-6). In experiment ID 1 and 2, we set one of the weights to 0 to evaluate the necessity of using the corresponding loss. We find that jointly optimizing the fMRI and image reconstruction loss is necessary to achieve optimal task performance. We adjust the $\gamma_F$ and $\gamma_i$ and find that setting in ID 6 where $\gamma_F = 0.25$ and $\gamma_i = 1.5$ yields the highest performance.

**Mask Ratio** In the FRL Phase 2, both the fMRI and image auto-encoders are based on MAE. The mask ratio of the input is an important setting for these models. We demonstrate the influence of the mask ratio setting in Table 2 (ID 4 and ID 7-10). We find that applying a higher mask ratio on fMRI data and a lower mask ratio on image data generally leads to better performance. LDM conditioned by the fMRI encoder which is trained with an fMRI mask ratio of 0.75 and image mask ratio of 0.5 leads to the highest reconstruction accuracy.

**Decoder Layers** Following [7, 51, 55], for both the fMRI and the image auto-encoder, we build asymmetric architectures where the decoder is much smaller than the encoder. In Table 2 (ID 11-13), we report the results of tuning decoder depth. We apply the same depth for the two decoders. We find that a moderate decoder depth of 6 produces optimal results. Neither a too shallow nor a too deep decoder improves reconstruction performance on GOD subject 3.

## 6 Discussion

The experimental findings indicate that, with the proposed fMRI representation learning framework and a pre-trained LDM, we can achieve a degree of visual reconstruction of human brain activities, substantially outperforming baselines. Nonetheless, our analysis also uncovers some limitations within our model. Primarily, our model appears to be slightly affected by a categorical bias issue. We hypothesize that this may stem from the inherent bias present in the dataset used to train the LDM. Additionally, while our model demonstrates proficiency in capturing high-level semantics, it sometimes fails to reconstruct specific details of an image. A plausible explanation could be the concurrent imagination of multiple objects by the participants during data collection, which inevitably results in a noisy fMRI feature. Contrasting with general image generation, which is typically characterized by a focus on diversity, visual decoding underscores the importance of consistency, thereby necessitating a reduction in bias during the generation process. As such, the exploration of techniques to minimize the influence of data bias, as well as methods to enhance the reconstruction of image details when generating images from fMRI data, would hold significant academic value and interest.

## 7 Conclusion

In this work, we propose a double-phase fMRI representation learning framework (FRL) with an LDM to reconstruct visual experiences from brain activities. The FRL denoises fMRI features by contrastive masked modeling in Phase 1. And in Phase 2 it learns to disentangle and attend to brain activation patterns most informative for visual decoding with the guidance of an image MAE. With the conditioning of the optimized fMRI feature learner, we show that an LDM generates images of high quality, largely outperforming the previous state-of-the-art. Extensive ablation studies further verify the effectiveness of each component that we propose.

## 8 Ethical Statements

We used preprocessed data from publicly available datasets. The fMRI data that we train with have been processed and do not contain any data that can be directly linked to the participants' identities. The collection procedure of the fMRI undergoes strict ethical review as stated in their original paper.

## 9 Acknowledgements

This work is funded by the CALCULUS project (European Research Council Advanced Grant H2020-ERC-2017-ADG 788506) and the Flanders AI Impuls Programme - FLAIR.

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

## Appendix

## A.1  Reconstruction Performance

We provide the performance of our model on BOLD5000 subjects 1, 2, 3, and 4 in Table A.1. Following previous work [7], all results are presented in 50-way-top-1 classification accuracy.

| BOLD 5000 | CSI 1 | CSI 2 | CSI 3 | CSI 4 |
|-----------|-------|-------|-------|-------|
| OURS | 25 | 18.69 | 16.14 | 18.98 |

Table A.1 Reconstruction performance of our model on BOLD5000 subject CSI 1-4, measured by 50-way-top-1 classification accuracy.

## A.2  Examples of Reconstructed Images

Figures A.2 and A.3 present images generated by our model using fMRI data from GOD and BOLD5000 datasets, respectively. We generated all images at a resolution of $256 \times 256 \times 3$ using 250 PLMS steps. More samples can be generated using our code base in the supplementary materials. The code will be open-sourced with the camera ready version of this paper.

## A.2.1 Reconstructed Images from GOD Dataset

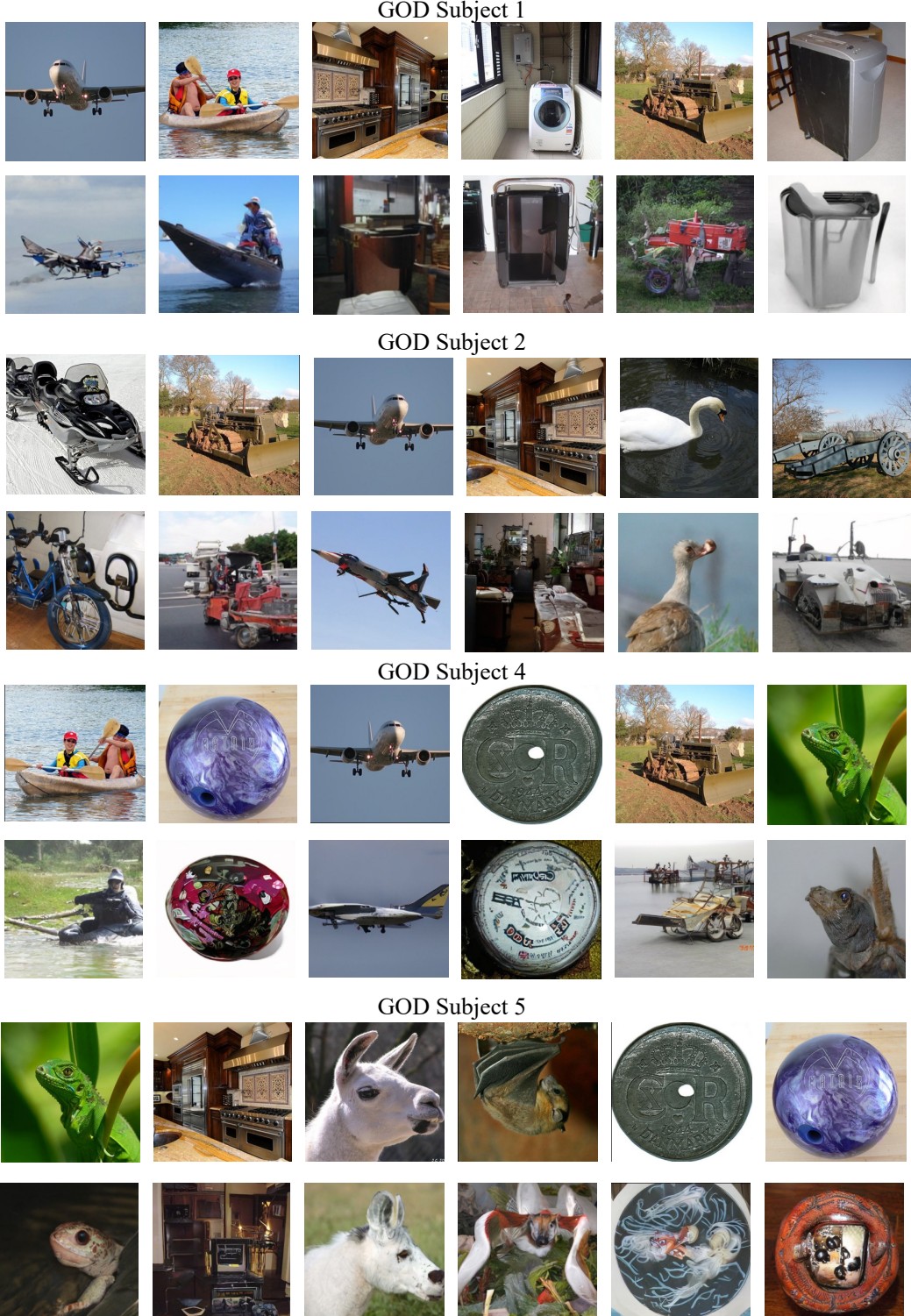

Figure A.2: Randomly selected reconstructed images from GOD subject 1, 2, 4 and 5. For each subject, the upper line shows the ground truth images while the lower line shows the reconstructed images by our method.

## A.2.2 Reconstructed Images from BOLD5000 Dataset

BOLD5000 CSI 1

BOLD5000 CSI 2

BOLD5000 CSI 3

BOLD5000 CSI 4

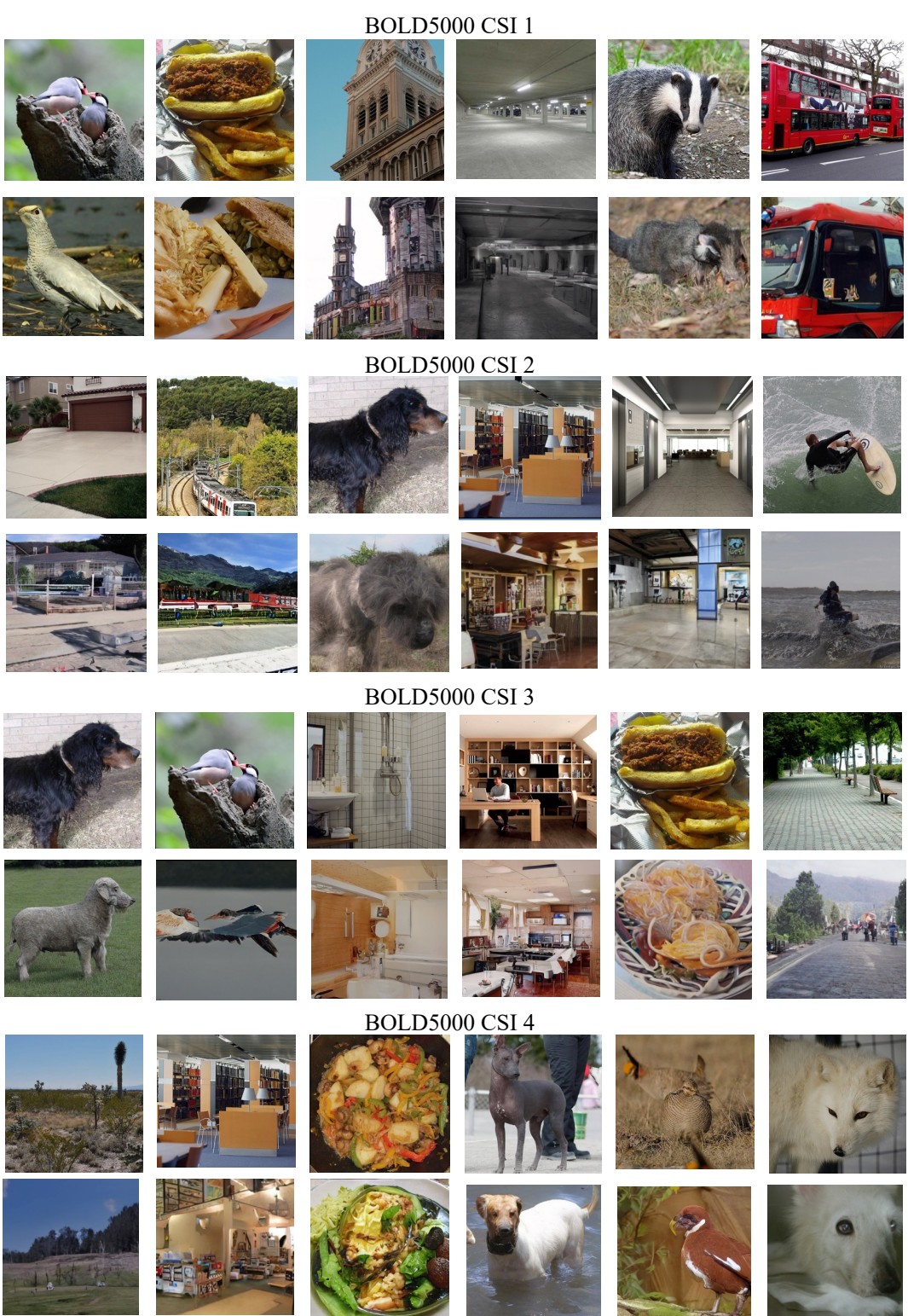

Figure A.3: Randomly selected reconstructed images from BOLD5000 CSI 1-4. For each subject, the upper line shows the ground truth images while the lower line shows the reconstructed images by our method.

### A.3 Evaluation Metrics

We use the common N-trial, n-way top-1 semantic classification as the main evaluation metrics. This evaluation method is summarized in the algorithm below:

---

**Algorithm 1** Iterative Reasoning Module

---

**Input:**

    pre-trained image classifier $F$, generated image $\hat{x}$, corresponding ground truth (GT) image $x_{gt}$

**Output:**

    success rate $sr \in [0, 1]$

    **for** $trail = 1$ to $N$ **do**

        $y_{gt} = F(x_{gt})$ get the prediction of GT image

        $pred = F(\hat{x})$ get the output probabilities of generated image

        $p = \{p_g, p_{y_1}, ..., p_{y_{n-1}}\}$ generate probabilities set contains $n - 1$ randomly selected from $pred$

        and $y_{gt}$

        Success if $\arg\min_{y} = y_{gt}$

    **end for**

    **return** $sr$ = number of success / N

---

