# OpenReview forum: "Contrast, Attend and Diffuse to Decode High-Resolution Images from Brain Activities"
_NeurIPS.cc/2023/Conference — NeurIPS 2023 poster_

### Official Review · Reviewer_5Cka · 2023-07-03

**Soundness:** 3 good
**Presentation:** 2 fair
**Contribution:** 3 good
**Rating:** 6
**Confidence:** 4

**Summary:**

The paper focuses on decoding visual stimuli from recorded fMRI activity. Proposed method initially pre-trains an fMRI feature learner (and essentially a signal denoiser) on unlabeled data using a contrastive training scheme resembling to masked autoencoders. Subsequently this feature encoder is finetuned by using guidance from an image auto-encoder to attend to signal patterns that are informative for reconstruction. Output of this fMRI feature learner is then used to condition a latent diffusion model to be able to visually reconstruct the stimuli in the image domain with high resolution.

**Strengths:**

The paper tackles a very interesting problem with nice, visually illustrated results. Proposed idea of performing neural signal guidance in the latent space such that a generative latent diffusion model can be used for high-resolution image synthesis is quite a neat approach to solve this problem.

**Weaknesses:**

There is no higher level justification or discussion present in the paper on why one would/should use fMRI for this problem (as opposed to other alternatives) towards building a non-invasive BMI. Moreover from a technical perspective, the paper only puts together several existing methods but some technical details are appearing to be mixed up in the descriptions. These should be corrected and some clarifications are still necessary in terms of the presentation of results.

**Questions:**

- Motivation of the study from a non-invasive brain-machine interface perspective is somewhat disputable, since the neurophysiological source signal considered here is fMRI. Can the authors extensively discuss and justify to the reader why their choice of fMRI (with low temporal resolution and longer temporal scales of responses, noisy nature, and hard-to-integrate experimental recording/design) would be a better candidate than other alternatives (e.g., EEG, or (partly-)invasive methods) to tackle this problem?

- Figure 2 with the "VQGAN" does not make sense. Starting from the last paragraph of page 5 and then in Figure 2, the paper talks about a VQGAN, whereas the LDM that the authors use should certainly be a pre-trained VQ-VAE with an encoder-decoder pair? Also based on the notation in Eq 9 and line 193, I presume the VQ-VAE encoder should be E_g rather than E_G on line 191?

- Description of Phase 1 has some ambiguities. Beginning of Sec 3.2 was understandable. Then line 140 starts to talk about an unmasked original "image" at some point, whereas the fMRI inputs were not yet denoted as "images" so far? The narrative can be made more consistent. Perhaps in Section 4.1 or 4.2, authors should also briefly describe the image-like fMRI input data representation scheme in a few words.

- Presentation of quantitative results are also ambigious. Figure 3 is not discussed in text, and achieved accuracies are not readable from the figure. Ablation studies from Section 5.2 are completely separated from the main quantitative result of the paper that is written in the Abstract: 39.34% in 50-way-top-1 semantic classification accuracy. In Section 5.1 or 5.2, it is not clear where this number really comes from? In general, the paper should revise its presentation of quantitative evaluations.

- Methodological comparisons to DC-LDM should be a bit more elaborated. Does this method only differ in the use of a contrastive pre-training phase, or are the encoder network architectures of the 24-layer FRL also different? Can the authors clarify the differences a bit more in depth on how significant their contributions are?

- Authors do not explicitly state this, but is the general idea of contrastively using a masked autoencoder scheme for denoising fMRI data a novel contribution of the paper? If not, can the authors refer to similar works and discuss that it is a powerful approach that is adopted for pre-training in their model?

Minor comments:
- It appears like in no occasion an fMRI conditioned reconstruction comes out directly identical to the ground truth, and hence it would be probably better/more realistic in Figure 2c to use a different airplane image as generated.
- What is the "CMA" block indicating in Figure 1? Is it the cross-attention module?
- In Figure 1, the dotted background pattern prevents the equation symbols/text to be easily read. Perhaps this figure text/equations can be re-generated in a better visible way.


**Limitations:**

Sufficiently addressed.

---

> ### Author Rebuttal · Authors · 2023-08-09
>
> Thanks very much for all the constructive feedback. We answer your questions and comments as follows. (Due to the rebuttal length limit, we summarize your questions.)
>
> Q1: Why fMRI is a better choice over alternatives such as EEG for this problem?
>
> A1: Primarily, fMRI offers significantly higher spatial resolution than EEG. This allows us to capture detailed activations within the visual cortex and related brain areas, a beneficial factor when reconstructing visual images from brain activity. Moreover, the presented stimuli in the dataset are all static images, making the fMRI’s limitation in temporal resolution less concerning. When working with static images, the prolonged temporal scales of fMRI responses provide ample time for the brain to process the stimuli and for the hemodynamic response to unfold. Therefore, considering both fMRI and EEG have inherent noises, fMRI's strength in spatial resolution and the nature of our stimuli make it a more fitting choice than EEG in this work. fMRI is also used in a lot of related work published in top-tier journals, such as Nature[1] and Science[2].
> (All the following work have been cited in our paper)
>
> [1] K. N. Kay, et al, “Identifying natural images from human brain activity,” Nature, vol. 452, pp. 352–355, 2008.
>
> [2] T. Horikawa et al., “Neural decoding of visual imagery during sleep,” Science, vol. 340, pp. 639 – 642, 2013.
>
> Q2: Why does Figure 2 reference a "VQGAN" when the LDM used a pre-trained VQ-VAE? Based on Eq 9, should the notation for the VQ-VAE encoder be E_g instead of E_G?
>
> A2: The VQGAN, at its core, can indeed be perceived as a VQ-VAE optimized with a GAN loss. However, as corroborated by LDM’s original paper [3], the model we adopted specifically leverages a VQGAN. In section 3.2 of the LDM's original paper, it's explicitly stated in the last paragraph, “This model can be interpreted as a VQGAN...”. The open-sourced code of the LDM provides further evidence of discriminative loss implementation.
> As for the notations, thanks for pointing this out, E_g should be used to denote the VQGAN encoder. We will correct the notation in line 193 in our revised manuscript.
>
> [3] R.Robin, et al. "High-resolution image synthesis with latent diffusion models." Proceedings of the IEEE/CVF Conference on CVPR. 2022. (ref. 24 in our paper)
>
> Q3: About making consistent references to fMRI inputs and adding a brief description of the fMRI representation scheme.
>
> A3: Thanks for the suggestions. As a kind of neuro-imaging, one sample of fMRI recordings can be called an image. But we acknowledge that it might be a little confusing to differentiate between the image of visual stimuli. We will follow your suggestions and unify the naming of fMRI data as fMRI samples in the revision.
> As for the image-like fMRI representations scheme, we divide the vectorized voxels into patches which will be subsequently transformed into embeddings using a 1D convolutional layer with a stride equal to the patch size (16). We will add these descriptions in Section 4.1.
>
> Q4: Is Figure 3 discussed in the text? How do the ablation studies from Section 5.2 relate to the main quantitative result mentioned in the Abstract? How is the 39.34% accuracy in the Abstract calculated?
>
> A4: Thanks for the questions.
> 1. We've indeed referenced Figure 3 within the text. Specifically, Figure 3[a,b,c] is elaborated upon in lines 244-248 and 250-252 of the Results Section 5.1.
> 2. Our ablation studies in Section 5.2 align with the main quantitative outcomes. Specifically, the concluding line of our ablation table (Table 2) indicates the hyperparameter setting yielding the 25.080 performance as illustrated in Figure 3[a].
> 3. The 39.34% mentioned in the abstract denotes a relative improvement, not an absolute accuracy. This is derived from the formula: (25.080−17.999)/17.999×100%=39.34%, comparing our model's accuracy (25.080) against the DC-lDM's accuracy (17.999). We will directly annotate Figure 3[a] with our accuracy value in the revision for better transparency.
>
> Q5: Does the method's distinction lie solely in the contrastive pre-training phase, or are there other differences in the FRL? How significant are the authors' contributions in relation to these differences?
>
> A5: Thanks for the questions. First, our model's Phase 1 indeed differs from the simple masked auto-encoder (MAE) utilized by DC-LDM by employing a novel double-contrastive MAE. Second, exclusive to our approach is Phase 2, characterized by our Cross Modality Guidance. As elaborated in section 3.2 and illustrated in Figure 1, this phase directs the fMRI encoder towards features crucial for image reconstruction—a facet absent in DC-LDM. Our ablation study, specifically in section 5.2.1, delves into the contributions of these mechanisms to the final performance.
> We'll ensure these distinctions are more pronounced in the revision to underscore the novelty of our approach.
>
> Q6. Is the general idea of contrastively using a masked autoencoder scheme for denoising fMRI data a novel contribution of the paper? If not, can the authors refer to similar works and discuss that it is a powerful approach that is adopted for pre-training in their model?
>
> A6: Indeed, our adoption of the contrastive masked autoencoder (MAE) scheme specifically for fMRI data denoising represents a novel contribution to neuroimaging. While the synergy of contrastive learning and MAEs has been explored in other domains, its application to fMRI is distinct, given the domain's inherent challenges like high dimensionality and pronounced noise levels. We've substantiated the efficacy of this approach in sections 5.1 (results) and 5.2 (ablation study). In light of your feedback, we'll emphasize this innovation more prominently in our revised manuscript.
>
> Response to minor comments: Yes, we will use a different plane image in Fig. 2. And the CMA block indeed refers to the cross-attention module. We will remove the dotted background in Fig. 1 in the revision.

---

> > ### Comment · Reviewer_5Cka · 2023-08-11
> > **response to the rebuttal**
> >
> > Thanks to the authors for their responses. I have examined all reviews and responses carefully. Several appreciated clarifications are made, and I'm more comfortable with this submission now.
> >
> > Justifications on choosing fMRI over alternatives is fair enough [A1]. But I still think that the motivating point-of-view should not be directly from the perspective of "building a non-invasive BMI", since I somehow can not see this application to yield a realistic BMI with an fMRI in the loop.
> >
> > All of the mentioned clarifications should be included in the revised manuscript. Some of the points that will be fixed in the revisions are quite important for the reader to follow the paper (e.g., sudden changes in terminology preferences). Figure 3a should be revised such that these bars are readable quantitatively. Numerical conclusions referring to residuals should be explicitly stated somewhere in the paper too [A4].
> >
> > Overall, I'd be happy to increase my score to the accept region, assuming presence of these revisions to higher clarity.

---

> > > ### Author Response · Authors · 2023-08-11
> > >
> > > Dear reviewer,
> > >
> > > First and foremost, we would like to express our deep gratitude for your time and  thoughtful feedback to improve this submission.
> > >
> > > We genuinely appreciate your insights and recognize the importance of the concerns you've raised. We understand your concerns regarding the potential application of fMRI-based decoders in non-invasive BMIs and will ensure that our revised manuscript offers a more balanced perspective.
> > >
> > > Your suggestions about the presentation such as terminology consistency, Figure 3a's readability, and the explicit statement of numerical conclusions are well-taken. We commit to addressing all of these in our revisions to provide clearer and more consistent content.
> > >
> > > In closing, your guidance has been invaluable in refining our work. We will diligently implement all your advice in the revised manuscript. Once again, thank you for taking the time and effort to provide constructive feedback on our manuscript and for considering our submission favorably.
> > >
> > > Warm regards,
> > >
> > > Authors of Submission6209

---

### Official Review · Reviewer_zrBS · 2023-07-03

**Soundness:** 3 good
**Presentation:** 3 good
**Contribution:** 4 excellent
**Rating:** 7
**Confidence:** 4

**Summary:**

The authors aim to decode the visual stimuli from neural responses by reverse-mapping the signals from functional MRI (fMRI) to the images the participants see while being scanned. The authors claimed to achieve this through a two-phase framework. In the first phase, they pre-train an fMRI feature learner inspired by mask auto-encoder (MAE) from unlabeled fMRI data. This phase tries to discern the common patterns and features in fMRI across participants responding the same stimuli. In the second phase, they further tune the fMRI feature learner with cross-modality correspondence from the visual stimuli, i.e., the images. This process is similar to the cross-modality guidance commonly used in vision language models. Finally, after the two phases of training, the fMRI feature learner is used to generate signals that serve as conditions for a latent diffusion model (LDM) for image generation.

The authors demonstrated that the final model is able to produce images from the participants’ fMRI signals such that the generated images well correspond — in category — to what the participants see. On the quantitative benchmarks, the proposed method outperform the baselines by a significant margin.

As much as I appreciate this work, I would like to demystify it a bit. In my opinion, it’s effectively performing two tasks sequentially: (1) fMRI signal classification into 50 or 100 classes, and (2) class-conditional image generation. The holistic framework and storytelling might have made it more impressive than it actually is — though admittedly it is still impressive.


**Strengths:**

1. “Mind reading” is a very bold topic. Generating the image in a participant’s mind from the fMRI recordings is arguably a reasonable way.

2. The design with three stages: contrastive pre-training, cross-modality fine-tuning and latent diffusion model is very reasonable and well conveyed.

3. The ablation studies are presented neatly.

**Weaknesses:**

1. The experimental results, though impressive, do not seem very comprehensive. For example, in Figure 3, qualitative results are shown for two datasets while bar plots are only displayed for one.

2. Also, I would recommend showing some intermediate results — for example, directly evaluating the fMRI representation accuracy by training a simple model to classify the learned fMRI representations after phase 2.


**Questions:**

1. Do you think it is reasonable to present the intermediate fMRI representation accuracy as described in Weaknesses 2? If I understand it correctly, you have $image_{real} \rightarrow (human) \rightarrow fMRI \rightarrow representation \rightarrow image_{gen}$, and you are showing $acc(classifier(image_{gen}), label(image_{real}))$. I wonder if you can also show $acc(classifier'(representation), label(image_{real}))$ and even $acc(classifier'(representation), classifier(image_{gen}))$.

**Limitations:**

Nothing came to my mind.

---

> ### Author Rebuttal · Authors · 2023-08-09
>
> Thank you very much for the valuable feedback and for appreciating our work.  Our responses to the weaknesses and questions are as follows.
>
> Weakness 1: In Figure 3, qualitative results are shown for two datasets while bar plots are only displayed for one.
>
> Answer 1: Thanks very much for the advice on the presentation of the results. We put the bar plot of BOLD5000 in the appendix due to the space limit. We will definitely put it in the main text during revision.
>
> Weakness 2 and Question 1: The reviewer would recommend showing some intermediate results — for example, directly evaluating the fMRI representation accuracy by training a simple model to classify the learned fMRI representations after phase 2. Do you think it is reasonable to present the intermediate fMRI representation accuracy?
>
> Answer 2:  Yes, it is very reasonable to evaluate the intermediate fMRI representation and we thank you for the insightful suggestion.  While assessing these representations is valuable, directly classifying them using our datasets is challenging. In BOLD5000, most image classes contain just one image. For GOD, with its 1200 images, test and training sets differ totally in class composition. Given fMRI's inherent noises and lessons from related works, training a precise classifier on these intermediate representations is very complex.
>
> However, prompted by your feedback, we've explored an alternative: cross-modality reconstruction in Phase 2, as illustrated in Figure 1. We believe that evaluating the masked auto-encoding results for images in Phase 2 can produce a fitting metric to assess intermediate representations. Specifically, for one image-fMRI sample pair, we mask 50% of the image.  The masked image is input into the image encoder, while the fMRI sample is directed to the fMRI encoder. The outputs from both encoders are combined and subsequently fed into the image decoder for reconstruction. Pearson’s correlation between the original and reconstructed images can also measure the quality of fMRI encoder's representations. With average correlations of 0.8971 for GOD subjects and 0.8703 for BOLD5000 subjects, we demonstrate the robustness of our intermediate fMRI representations.

---

> > ### Comment · Reviewer_zrBS · 2023-08-18
> > **Response to Rebuttal**
> >
> > I have read the rebuttal by the authors. They well explained the challenge in training a separate classifier for classifying the intermediate representations. I do not have any further concerns, and decide to maintain my rating at 7.

---

### Official Review · Reviewer_BiRm · 2023-07-05

**Soundness:** 3 good
**Presentation:** 3 good
**Contribution:** 3 good
**Rating:** 6
**Confidence:** 4

**Summary:**

This paper proposed to decode visual stimuli from neural responses recorded by fMRI. First, it pretrains an fMRI feature learner with a proposed Double-contrastive Mask Auto-encoder to learn denoised representations. Second, it tunes the feature learner to attend to neural activation patterns most informative for visual reconstruction with guidance from an image auto-encoder.


**Strengths:**

(1) The method is straightforward and easy to understand.
(2) The related work discussion seems comprehensive.


**Weaknesses:**

(1) The selection of baseline methods for comparison appears to be insufficient, thus hindering the ability to effectively demonstrate the effectiveness of the proposed method. To enhance the evaluation, it would be beneficial for the authors to include a comparison with the methods mentioned in the related work section.

(2) Another crucial factor that warrants consideration is the quality of the fMRI data, as it can significantly impact the performance of the model. It would be valuable if the authors could provide some ablation studies pertaining to this aspect as well.

(3) The evaluation metric solely employed is n-way top-k accuracy, which could be supplemented with additional evaluation metrics to provide a more comprehensive assessment. Metrics such as FID (Fréchet Inception Distance), SSIM (Structural Similarity Index), MSE (Mean Squared Error), among others, would be advantageous for a more qualitative evaluation.

(4) Since human data is used in the study, the authors can discuss some potential ethical concerns.


**Questions:**

Please see the comments above.


**Limitations:**

The limitation is discussed in Section 6, however, since human data is used in the study, the authors can discuss some potential ethical concerns.

---

> ### Author Rebuttal · Authors · 2023-08-09
>
> Thanks very much for all the constructive feedback. We answer your questions as follows. (Due to the rebuttal length limit, we might summarize some of your questions.)
>
> Q1: About the selection of baselines and consideration of other models cited in related work as baselines.
>
> A1: Thanks for your advice on baseline selection. We have three points to clarify. First, we follow previous work DC-LDM’s selection for baselines to ensure a fair evaluation.  DC-LDM is the previous SOTA model which has been peer-reviewed and accepted by a top-tier conference (CVPR 2023). Compared against DC-LDM, our model shows substantial relative improvements of 39.04%, as depicted in the Results Section 5.1.  Second, we described GAN-based, Diffusion-based, and VAE-based methods in the related work. All three categories have their corresponding model in the baselines, IC-GAN for GAN, DC-LDM for diffusion, and SS-AE for VAE. Third, we did not include traditional regression-based methods, since their performances are not comparable to the latest deep learning based methods as we introduce in the related work Section 2.1.
>
>
> Q2: About the impact of fMRI data quality and ablation studies pertaining to this aspect.
>
> A2: Thank you for emphasizing the significance of fMRI data quality. We have three points to explain regarding your questions.
> First, our method is explicitly tailored to fMRI’s noisy nature. We employed a contrastive masked autoencoder to derive denoised fMRI representations. The effectiveness of this approach is evident in our superior task performance.
> Moreover, we indeed have some related ablation studies with masking on fMRI, as reported in Table 1 and Table 2. In masking, we randomly set part of the fMRI representation to zero which is also a kind of quality impairing. Our optimal task performance is achieved by an fMRI encoder trained with 75% masked fMRI inputs, proving our model’s robustness to fMRI quality.
> Last, it's pertinent to note that non-invasive neuroimaging, including fMRI, inevitably comes with inherent noises. Despite this, the BOLD5000 and GOD datasets, utilized in our study, have been widely acknowledged in brain decoding tasks, underscoring their quality. For example, BOLD5000 was evaluated using MRIQC [1], yielding a Signal-to-noise ratio of 5.157, attesting to its reliability.
>
> [1] Chang, Nadine, et al. "BOLD5000, a public fMRI dataset while viewing 5000 visual images." Scientific data 6.1 (2019): 49. (ref. 46 in our paper)
>
> Q3:  About supplementing additional evaluation metrics  to provide a more comprehensive assessment.
>
> A3:  Thanks for your suggestions. We follow the previous SOTA DC-LDM for the evaluation setting. We further evaluate our model using MSE and SSIM based on your suggestions. We report the comparisons of our model with DC-LDM. After averaging across all subjects, our model achieves an SSIM of 52.48 and MSE of 49.59, significantly outperforming DC-LDM’s 51.85 SSIM and 51.38 MSE on the BOLD5000 dataset (p<0.01 with paired t-test for all the significant results).  We will definitely supplement these metrics in the revised manuscript.
>
> Q4:  Since human data is used in the study, the authors can discuss some potential ethical concerns.
>
> A4: Thanks very much for the suggestions. We used preprocessed data from publicly available datasets. The fMRI data that we train with have been processed and do not contain any data that can be directly linked to the participants’ identities. The collection procedure of the fMRI undergoes strict ethical review as stated in their original paper. We will definitely add a section of the ethical statement at the end of the paper.

---

> > ### Comment · Reviewer_BiRm · 2023-08-18
> > **Thank you for the rebuttal**
> >
> > Thanks the authors for the rebuttal. My concerns have been addressed and I have changed my score to weak accept.

---

### Official Review · Reviewer_zH9U · 2023-07-13

**Soundness:** 3 good
**Presentation:** 3 good
**Contribution:** 3 good
**Rating:** 6
**Confidence:** 3

**Summary:**

The paper proposes an approach for decoding visual stimuli from neural responses (fMRI images). The rationale behind the proposed approach lies in the difficulty of learning the complex relationship between a stimuli and the neural responses to it, and the noisy nature of fMRI images.

The authors propose a multiple phase approach comprising a contrastive learnign based method for learning denoised representations of fMRI brain activities (a new DC-MAE method is proposed), a feature learner that combines previsous representation space with image representation through  cross attention and a diffusiuon model to reconstruct image stimuli from brain activities which is conditioned on the opuput of the feature learner.


**Strengths:**

The approcah looks like a smart agregation of state of the art components that allows reaching high top-k accuracy (the chosen metric).

The paper is easy to read and the methodological choices are well motivated.

The method is validated by top k accuracy and achieves significantly better results than the baselinesit is compared to.

A thourough ablation study shows the impact of the various components and of the hyperparameters of the method.


**Weaknesses:**

Few aspects of the method are not enough detailed (see questions)



**Questions:**


To which space does v_i belongs, is is a time series ?  I don’t understand the use of 1D conclutional models that maps v_i^m_1  and v_i^m_2 into embeddings.

In Phase2 are all models (E_F, E_I, D_F, D_I) retrained or are only the cross attention layers trained ?

The way the latent diffusion model is conditionned seems new. Hwo does this compare to previous attempts of designing conditional latent diffusion models ?

Concerning metrics it is said in section 4.3 « We employ the pre-trained ImageNet-1K classifier [50] as a semantic correctness evaluator. » wht does this mean ?

Concerning the top-k accuracy experiments did the authors use a statistical test to attest the significativity of the result ?

**Limitations:**

limitations are addressed

---

> ### Author Rebuttal · Authors · 2023-08-09
>
> Thanks very much for all the constructive feedback and for appreciating our work. We answer your questions as follows.
>
> Q1: To which space does v_i belong, is it a time series? Why  1D convolutional model is used to map v_i^m_1 and v_i^m_2 into embeddings?
>
> A1: Thank you for your question regarding the nature of v_i and the usage of 1D convolutional models in our study.
> In our work, v_i denotes the fMRI signal when the subject is viewing a picture. Importantly, it is a 1D vector, not a time series, as we have averaged the data across the time dimension. This results in a spatial pattern of fMRI signal over the visual cortex for each picture viewed by the subject. We then employ a 1D convolutional model to transform this 1D spatial pattern of the fMRI signal, v_i, into an embedding. We will add these details during revision.
>
> Q2: In Phase 2 are all models (E_F, E_I, D_F, D_I) retrained, or are only the cross attention layers trained?
>
> A2: All models (E_F, E_I, D_F, D_I) are retrained, as described in line 164.
>
> Q3: The way the latent diffusion model is conditioned seems new. How does this compare to previous attempts at designing conditional latent diffusion models?
>
> A3: Thanks for your question. We apply this type of conditioning on LDM given the noisy nature of fMRI data. We adopt both cross-attention conditioning and time-step conditioning. Time-step conditioning strengthens the overall conditioning effect. Cross-attention conditioning helps to create a stronger condition. This double conditioning is inspired by the conditioning methods in [1] and [2].
>
> [1] R. Robin, et al. High-resolution image synthesis with latent diffusion models. In Proceedings of the IEEE/CVF Conference on  CVPR, pages 10684–10695.   (ref. 24 in our paper)
>
> [2] P. Dhariwal et al.“Diffusion models beat gans on image synthesis,” Advances in NeurIPS, vol. 34, pp. 8780–8794, 2021. (ref. 44 in our paper)
>
>
> Q4: What does this  “We employ the pre-trained ImageNet-1K classifier [50] as a semantic correctness evaluator”  mean?
>
> A4: Thanks for your question. We follow the previous work using n-way top-k accuracy to evaluate our model.  A detailed explanation of the computation for this metric can be found in Algorithm 1 provided in the appendix.  Regarding the statement,  “We employ the pre-trained ImageNet-1K classifier as semantic correctness evaluator”, we mean that we use the pre-trained ImageNet-1k classifier to classify both ground truth images and the generated images. These predicted classes are then utilized in computing the final metrics as explained in Algorithm 1.
>
> Q5: Concerning the top-k accuracy experiments did the authors use a statistical test to attest the significance of the result?
>
> A5: Yes, the improvement of our model over the baselines is significant, including the previous state-of-the-art DC-LDM. All the significant results have p-value < 0.01 with a paired t-test. We will further clarify them in the revision.

---

> > ### Comment · Reviewer_zH9U · 2023-08-16
> > **Response to the rebuttal**
> >
> > Thanks to the authors for their responses. I read all reviews and the answers carefully.
> >
> > Many clarifications and additional results have been obtained that strengthen the paper. Yet I am not fully satisfied with the answer about the nature of v_i, and more generally I still feel the description of the method lacks details.
> >
> > I still don’t understand why a 1D convolutional layer is used, why are the data averaged across the time dimension, and if so what is the axis along which a 1D convolutional layer is used? Also, what do you mean by 1D spatial pattern?
> > I am not an expert in neuroscience and i likely lack some background knowledge to understand the processing you describe which is maybe standard in the neuroscience field.
> >
> > Yet I believe providing more details would help any machine learning reader to understand the paper better even if he/she is a non expert in the field of neuroscience.

---

> > > ### Author Response · Authors · 2023-08-17
> > >
> > > Dear reviewer,
> > >
> > > Thank you very much for taking the time to provide thoughtful feedback on our manuscript and read the rebuttals. We here elaborate further on some standard processing pipelines in fMRI for your convenience.
> > >
> > > To answer your questions, we need first briefly introduce some background of fMRI. Thanks for your patience.
> > >
> > > At a basic level, fMRI can be thought of as capturing a series of 3D images of the brain at fixed intervals. When a participant is presented with an image for $t_p$ seconds and the repetition time is $t_r$ seconds, we obtain a sequence of $t_p / t_r$ fMRI samples after each presentation. Given an fMRI sample size of [$x, y, z$], the resulting sequence is of shape [$t_p / t_r, x, y, z$]. The first axis (with shape $t_p / t_r$) represents the temporal dimension or the time dimension. The other three axes refer to the spatial dimensions of the acquired data in 3D space.
> > >
> > > (1)  Why are the data averaged across the time dimension?
> > >
> > > fMRI measures hemodynamic response, which is the change in blood flow and blood oxygenation in the brain that corresponds to neural activity. Given that the hemodynamic response is slow and can span several seconds, individual fMRI volumes (taken every $t_r$ seconds) might capture only parts of this response. By averaging, we obtain a more stable and comprehensive representation of the neural activity elicited by the stimulus, reducing the effects of transient fluctuations and noise. It is a common operation in fMRI preprocessing.
> > >
> > > (2)  Why a 1D convolutional layer is used and along what axis?
> > >
> > > After obtaining the averaged representation, we flatten the 3D tensor ([$x, y, z$]) into a 1D tensor, leading to the shape $x \times y \times z$ which is a common operation in fMRI processing [1]. With the flattened tensor, we then apply a 1D convolution. Given the spatial redundancy inherent in fMRI data, adjacent voxels are often found to display similar magnitudes as we describe in Section 3.1 line 109. The convolution operation is particularly suited for our data as it helps in aggregating localized information.
> > >
> > > (3) What is the meaning of spatial pattern?
> > >
> > > Simplistically, a neural activation pattern reveals how different brain regions activate in response to a stimulus. The term "spatial pattern" refers to the spatial distribution of these activation patterns, as also used in other work [2,3]. Given the spatial redundancy in fMRI, we apply a convolution layer to effectively aggregate local information and learn the spatial pattern.
> > >
> > > We genuinely appreciate your insights and guidance. We'll ensure to weave these explanations into the revised manuscript, catering to readers from diverse backgrounds. Should you have any further questions or require additional clarifications, please do not hesitate to propose them.
> > >
> > > Warm regards,
> > >
> > > Authors of Submission6209
> > >
> > > [1] Jang, Hojin, et al. "Task-specific feature extraction and classification of fMRI volumes using a deep neural network initialized with a deep belief network: Evaluation using sensorimotor tasks." NeuroImage 145 (2017): 314-328.
> > >
> > > [2] Hsieh, P-J., Ed Vul, and Nancy Kanwisher. "Recognition alters the *spatial pattern* of fMRI activation in early retinotopic cortex." Journal of Neurophysiology 103.3 (2010): 1501-1507.
> > >
> > > [3] Williams, Mark A., Sabin Dang, and Nancy G. Kanwisher. "Only some *spatial patterns* of fMRI response are read out in task performance." Nature Neuroscience 10.6 (2007): 685-686.

---

> > > > ### Comment · Reviewer_zH9U · 2023-08-19
> > > > **response**
> > > >
> > > > Dear authors
> > > >
> > > > I thank you very much for providing all this detailed information that is mandatory in the final version in my opinion.
> > > >
> > > > Indeed I would not have imagined such a reshaping before using a 1D convolutional layer as it does not capture all the neighboring information. I would have rather used a 3D convolutional layer (or at least 2D convolutional layer) to take into account all neighboring voxels. Is this common practice to use such a process?

---

> > > > > ### Author Response · Authors · 2023-08-20
> > > > >
> > > > > Dear reviewer,
> > > > >
> > > > > Thank you for your interest and continual feedback. We appreciate the opportunity to further clarify the details of fMRI preprocessing pipeline. Regarding your questions:
> > > > >
> > > > > Q1: Is reshaping into a 1D vector a common practice to use?
> > > > >
> > > > > A1: Yes, indeed. A significant part of the neuroscience literature has worked with reshaped 1D fMRI data [1-5], including our baselines to be compared. By following these work to use a 1D representation, we ensure a direct, consistent and fair comparison.
> > > > >
> > > > > Q2: Why do the reshaping but not use a 3D convolution?
> > > > >
> > > > > A2:  We definitely agree that 3D convolution is an interesting approach to explore in our future work. But in our current work, 1D convolution is also a reasonable solution. The reasons are as follows.  The 3D structure of fMRI signals record not only activities of the cortical surface (grey matter),  but also the subcortical areas and the white matter areas which are parts of the brain that are located below the cerebral cortex. However, for the visual reconstruction task, we are mainly concerned with the information from the cortical **surface**, which has been proven to be more relevant for conscious visual perception and representation in the brain [6, 7].  As a common operation taken in [1-5], the subcortical areas are removed during preprocessing which actually disrupts the 3D continuity of the data. With this preprocessing, 3D convolution would attempt to learn from disjointed spatial patterns and could potentially influence the meaningful signals from the surface regions. Moreover, 3D convolution increases the number of trainable parameters in the model than 1D convolution. With the limited data of the fMRI, possible overfitting is a concern. By using 1D convolution on the flattened data, we could limit model complexity while still retaining enough information to accurately decode the stimulus. We will diligently weave these detailed clarifications in the revised paper.
> > > > >
> > > > > We deeply appreciate your insights and guidance. Should you have any further questions or require additional clarifications, please do not hesitate to propose them.
> > > > >
> > > > >
> > > > > Warm regards,
> > > > >
> > > > > Authors of Submission6209
> > > > >
> > > > > [1] Z. Ren, J. Li, X. Xue, X. Li, F. Yang, Z. Jiao, and X. Gao, “Reconstructing seen image from
> > > > > brain activity by visually-guided cognitive representation and adversarial learning,” NeuroImage,
> > > > > vol. 228, 2021 (ref.4  in the paper)
> > > > >
> > > > > [2] Y. Takagi and S. Nishimoto, “High-resolution image reconstruction with latent diffusion models
> > > > > from human brain activity,” in CVPR 2023  (ref.5  in the paper)
> > > > >
> > > > > [3] Z. Chen, J. Qing, T. Xiang, W. L. Yue, and J. H. Zhou, “Seeing beyond the brain: Masked
> > > > > modeling conditioned diffusion model for human vision decoding,” in CVPR 2023 (ref.6  in the paper)
> > > > >
> > > > > [4] M. Mozafari, L. Reddy, and R. van Rullen, “Reconstructing natural scenes from fmri patterns
> > > > > using bigbigan,” IJCNN 2020  pp. 1–8. (ref.7  in the paper)
> > > > >
> > > > > [5] F. Ozcelik and R. VanRullen, “Brain-diffuser: Natural scene reconstruction from fmri signals
> > > > > using generative latent diffusion,” arXiv preprint arXiv:2303.05334, 2023. (ref.22  in the paper)
> > > > >
> > > > > [6]  Cecere, R., Bertini, C., & Ladavas, E. (2013). Differential contribution of cortical and subcortical visual pathways to the implicit processing of emotional faces: a tDCS study. Journal of Neuroscience, 33(15), 6469-6475.
> > > > >
> > > > > [7] National Institute of Neurological Disorders and Stroke. Brain Basics: The Life and Death of a Neuron.(https://www.ninds.nih.gov/health-information/public-education/brain-basics/brain-basics-life-and-death-neuron) Accessed 3/19/2023.

---

### Official Review · Reviewer_ir1U · 2023-07-25

**Soundness:** 3 good
**Presentation:** 3 good
**Contribution:** 3 good
**Rating:** 5
**Confidence:** 5

**Summary:**

The paper propose a novel two-phase fMRI representation learning method to encode the visual stimuli from neural responses. It is a significant challenging task due to noisy fMRI signals and complex intricate pattern of brain visual representation. The proposed two-phase can reconstruct image stimuli from brain activities.

**Strengths:**

1.The overall framework is novel and interesting. The use of double-contrastive mask auto-encoder and image-guided auto-encoder are novelty.
2.The introduction is clear.
3.The superiority experimental results are obtained.

**Weaknesses:**

1.The motivation of two phase is confused. Why two phase representation learning is needed? And what is the difference between the two phase?
2. How the contrastive MAE works? It is not clear
3.The model may be too big? How the efficiency?
The experiments are weak. How to set the mask? The model’s size? More quantification is needed.

**Questions:**

see above

**Limitations:**

the proposed method is too complex to follow.

---

> ### Author Rebuttal · Authors · 2023-08-09
>
> Thanks very much for all the constructive feedback. We answer your questions as follows.
>
> Q1: Why is two-phase representation learning needed? What is the difference between the two phases?
>
> A1: The two-phase design stems from the unique characteristics and challenges posed by fMRI data in the context of visual reconstruction, as stated in Section 3.1, lines 104-122.  For your convenience, we elucidate the motivation of and difference between the two phases as follows.
>
> First,  fMRI data is spatially redundant. Adjacent voxels tend to display similar magnitudes.  The mask and predict methodology in Phase 1 is designed to help the fMRI encoder learn the underlying structure of the input brain data, crucial for understanding brain dynamics.
>
> Second, fMRI data is noisy and subject to individual biological variances. Masked autoencoding in Phase 1 helps suppress noise. Optimization of the contrastive loss further discerns common patterns of brain activities over individual variances.
>
> Third, the process by which a visual stimulus arouses a neural response involves multiple stages of neural processing. The resulting fMRI signal is a highly convolved representation of these distinct stages.  So we design Phase 2 with cross-modality guidance, which aims to instruct the fMRI encoder to capture the most informative signals from the convolved fMRI data for visual reconstruction.
>
> In essence, our two-phase work synergistically: Phase 1 focuses on data denoising and structure understanding, while Phase 2 emphasizes extracting information crucial for visual reconstruction. Collectively, they ensure precise differentiation between cognitive states, which is paramount for brain decoding.
>
> Q2: How does the contrastive MAE work?
>
> A2:  Thank you for seeking clarity on the workings of the contrastive MAE. You can refer to Figure 1 and Section 3.2 Phase for details of contrastive MAE works.
> To elucidate step by step:
>
> 1) For each fMRI input sample, we generate two distinct masked versions.
>
> 2) These masked versions are then processed through the fMRI autoencoder, resulting in two reconstructed samples.
>
> 3) These two reconstructed samples are treated as a pair of positive samples.
>
> 4) Furthermore, the association between each reconstructed sample and the original unmasked sample also forms a pair of positive samples.
>
> 5) Reconstructed samples from other fMRI inputs in the same batch serve as negative samples.
>
> 6) With these established positive and negative pairs, we then optimize the contrastive loss as detailed in equations (1-3).
>
> Q3: What is the size and efficiency of the model? How to set the mask?
>
> A3:  Size and Efficiency of Model: We take training on the GOD dataset as an example.
> For the fMRI representation learning model, we train Phase 1 for 150 epochs and Phase 2 for 60 epochs on one Nvidia A100 GPU. The two phases in total take about 12 hours. After the two phases, we only save the checkpoint of the fMRI encoder which has 15.16M parameters.
> For the diffusion model, we use the pre-trained label-to-image latent diffusion model. The model has 401.32M parameters, but during finetuning, we only tune the weights in the cross-attention layers and the fMRI encoder which has 17.4M parameters in total. We use one Nvidia-V100 GPU to finetune the model for 500 epochs and it takes around 20 hours.
>
> Setting of Mask: We have conducted experiments to study the effects of masking ratio on reconstruction performance and the results are detailed in the ablation Table 1 and 2. We also include the mask setting to achieve the best reconstruction performance on the last line of Table 2, where the fMRI mask ratio is 0.75 and the image mask ratio is 0.5.
> We will definitely further clarify these quantifications in the revision.

---

> > ### Comment · Area_Chair_DcMS · 2023-08-22
> > **Response to rebuttal**
> >
> > Dear Authors,
> >
> > Thank you for answering the points that were raised, your response will be taken into account.
> >
> > Best,
> >
> > Your Area Chair

---

### Decision · Program_Chairs · 2023-09-21

**Decision:**

Accept (poster)

**Comment:**

This paper present an approach for reconstructing images from fMRI activity. The approach relies on a framework for learning representations from fMRI that conditions images generated by a diffusion model. The results appear strong and the approach is different from existing reconstruction approaches and would be useful to discuss at NeurIPS. The authors are expected to include their new materials and clarifications in their paper, along with modifying the abstract to make the improvement in performance more clear.